# Analysis of adherence to a healthy lifestyle among vegetarian and non-vegetarian Peruvian university students: A cross-sectional survey

Jacksaint Saintila[1]*, Ricardo Rojas-Humpire[2], Edda E. Newball-Noriega[2], Salomón Huancahuire-Vega[2], Felipe L. Ignacio-Cconchoy[1], Yaquelin E. Calizaya-Milla[3]

1 Doctorado en Nutrición y Alimentos, Escuela de Posgrado, Universidad San Ignacio De Loyola, Lima, Perú, 2 Escuela de Medicina Humana, Universidad Peruana Unión, Lima, Perú, 3 Research Group for Nutrition and Lifestyle, School of Human Nutrition, Universidad Peruana Unión, Lima, Perú

* jacksaintsaintila@gmail.com

## Abstract

### Background

Evidence shows that a healthy lifestyle can promote physical and mental well-being in the general population. However, there are few studies that assess the adherence to a healthy lifestyle in vegetarian and non-vegetarian university students. The purpose of this study was to evaluate the differences in adherence to a healthy lifestyle between vegetarian and non-vegetarian university students in Peru.

### Methods

A cross-sectional study was carried out considering data from 6,846 students selected by non-probabilistic convenience sampling. The Diet and Healthy Lifestyle Scale (DEVS), the Peruvian validation of the Vegetarian Lifestyle Index (VLI), was used. In addition, sociodemographic and anthropometric data such as weight and height were collected. Body mass index (BMI) was also calculated.

### Results

Semi-vegetarian and vegetarian students had a high healthy lifestyle score compared to non-vegetarians. In addition, vegetarian diets showed a significantly higher proportion among students with a lower BMI (normal and underweight). Students with excess body weight (overweight and obesity) were less likely to report healthy lifestyle. In the overall population analyzed, it was observed that the levels of health and lifestyle behaviors, such as daily exercise and sunlight exposure, were mostly moderate and low. Additionally, sweets intake was high, while healthy food consumption such as fruits, vegetables, legumes, and whole grains was low.

**Data Availability Statement:** The datasets generated to support the findings of this study are not publicly accessible due to ethical and legal constraints imposed by the Research Ethics Committee of the Universidad Peruana Unión. However, interested parties may request access to these datasets via email to the Ethics Committee at etica@upeu.edu.pe or to the university's General Research Directorate at director.dgi@upeu.edu.pe.

**Funding:** The study was financed by the Universidad Peruana Unión, Peru (Resolución N° 2556-2022/UPeU-CU). The funders had no role in study design, data collection and analysis, decision to publish, or preparation of the manuscript.

**Competing interests:** The authors have declared that no competing interests exist.

## Conclusion

The current findings show that although vegetarians had better adherence to a healthy lifestyle, interventions in the university setting are needed to improve healthy lifestyle in university students.

## Introduction

Vegetarian dietary patterns are predominantly characterized by the consumption of minimally processed plant foods, such as fruits, vegetables, whole grains, nuts, legumes, and seeds [1]. Currently, there is no universally accepted definition for vegetarian diets. However, vegetarianism generally does not include foods such as red meat, poultry, although it sometimes includes a limited amount of fish, eggs, dairy products, and honey [2]. For example, vegans are stricter, because they exclude all animal foods and foods of animal origin (e.g., honey) [2]. Lacto-ovo-vegetarians include dairy, eggs, and honey, while semi-vegetarians and pesco-vegetarians allow a limited amount of meat and/or fish in their diet [3]. Epidemiological evidence has associated vegetarian diets with a lower risk of non-communicable diseases such as type 2 diabetes mellitus, cardiovascular diseases, hypertension, and different types of cancer and a better quality of life [4–6]. The preventive role of vegetarian diets and their role in the treatment of chronic conditions could be due to the composition and nutritional benefits of the foods that make them because they are rich in dietary fiber, complex carbohydrates, vitamins and minerals, antioxidants, and other bioactive elements [7].

When vegetarian diets are inadequately planned, they can present a risk of deficiency of critical nutrients such as protein, omega-3 fatty acids, vitamins D and B-12, calcium, zinc, and iron [8]. Deficiency of these nutrients can negatively impact the health of vegetarians, particularly resulting in protein depletion, increased risk of neurological diseases, depression, non-communicable diseases, and long-term anemia [9]. In fact, findings from a recent study show that an unhealthy plant-based diet is associated with an increased risk of coronary heart disease [10]. However, according to the Academy of Nutrition and Dietetics, in an official position document, properly planned vegetarian diets are appropriate at all stages of the life cycle, including pregnancy, infancy, childhood, and adolescence [1]. While vegetarian dietary patterns are a healthy option, careful planning is necessary to ensure that they are appropriate [11]. Consequently, evaluation of vegetarian diets among university students and the general population is necessary to promote nutrition education and improve dietary practices among vegetarians and non-vegetarians. This could be essentially important for societies where there is a high prevalence of noncommunicable diseases, and few studies related to a healthy lifestyle [12].

Although the evaluation of vegetarian diets among university students and the general population is important in promoting healthy dietary practices, it is equally relevant to examine other dietary patterns that have been shown to be beneficial to health. One example is the Mediterranean diet, recognized for its nutritional balance and its positive effects on the prevention of non-communicable diseases. Recent studies have explored the adherence to the Mediterranean diet in different populations including university students [13–15]. These studies highlight the importance of balanced diet patterns, characterized by a high intake of fruits, vegetables, whole grains, and healthy fats, like some aspects of well-planned vegetarian diets [16]. These dietary patterns offer a broader perspective on how various dietary practices contribute to overall health and well-being, especially in contexts of high prevalence of non-

communicable diseases and crises such as the COVID-19 pandemic and adverse economic situations.

Adherence to a healthy lifestyle refers to the practices that a person adopts that can promote physical and mental well-being. These practices generally include a healthy diet, adequate water consumption, regular physical exercise, sunlight exposure, stress management, adequate sleep, and avoidance of harmful habits, such as smoking or excessive alcohol consumption [17]. Adherence to a healthy lifestyle in a university setting can be difficult for students due to the conditions and characteristics of the environment [18]. In addition, the transition from high school to university constitutes one of the most critical periods for young adults because they face several challenges related to risky behaviors, including changes in diet quality, and eating habits, physical inactivity, and consumption of harmful substances such as tobacco and alcohol [19], which negatively affect well-being and increase the risk of morbidity and mortality [20]. Likewise, the behavioral changes observed in young adults in the university environment imply a risk of weight gain, especially during the first year of university [21, 22]. These inappropriate behaviors may be associated with increased independence in food choice, low food acquisition capacity, and exposure to new social environments and food cultures [23]. The university stage, therefore, is a vital period to foster a culture of new healthy lifestyle practices, including healthy dietary patterns.

There is little information on the eating behaviors of Peruvian university students. According to the Ministry of Health, people aged 15 years and older, which also includes the university population, do not consume the portions of fruits and vegetables recommended by the World Health Organization (WHO) [24]. Generally, the transition from home to university life is associated with unhealthy lifestyle habits, including sedentary behaviors [25] and inadequate dietary habits, such as skipping breakfast, high consumption of saturated fats and added sugars [23]. On the other hand, vegetarian diets are gaining popularity, especially among young people, due to their perception as a healthier, sustainable, and ethical option compared to traditional meat-based diets. In addition, there is growing interest in understanding how dietary choices, particularly the adoption of a vegetarian diet, affect overall health and well-being. Studying these differences can provide valuable information about the benefits and potential challenges of following a vegetarian diet compared to a non-vegetarian diet, especially at a stage of life where long-term eating and lifestyle patterns are being established. In that sense, the comparison between vegetarians and non-vegetarians is critical to understand not only the direct health effects of these diets, but also to inform individuals, educational institutions, and policy makers on how best to support the health and well-being of young people in a context of dietary diversity. Therefore, the objective of this study was to assess the differences in adherence to a healthy lifestyle between vegetarian and non-vegetarian university students.

## Materials and methods

### Study design and population

This cross-sectional study was conducted in a private university in Peru with campuses in the three regions (coast, jungle, and highlands) of the country. The data was collected during the enrollment process to enter academic cycle I, between the months of February and March 2022. Participants were recruited by non-probability convenience sampling. Students were invited to participate in this study through a virtual survey accessible from the university's academic portal. Students had access to the survey when they entered the virtual classroom to follow their classes online or to carry out academic activities. The survey consisted of 28 items and included the following: (a) Diet and Healthy Lifestyle Scale (DEVS) to assess adherence to

a healthy lifestyle; (b) sociodemographic data (e.g., age, sex, origin, place of residence, parents' level of education, among others); and (c) self-reported information on body weight and height to subsequently calculate body mass index (BMI). To minimize bias or error due to self-reporting, we emphasized to participants the importance of providing accurate information, assuring them of confidentiality, and explaining how their data would contribute significantly to the research. In addition, we provide detailed and standardized instructions on how to measure height and weight. Students aged 18 to 29 years were included. A total of 8,769 students agreed to voluntarily participate in the study. However, students who did not meet the age criteria, those enrolled in postgraduate programs, who gave inadequate answers to certain survey questions, and those who did not give their consent were excluded. After applying the eligibility criteria, the final study sample size was 6,846 undergraduate students.

## Ethical aspects

Before starting the survey, participants received information on the objective of the study, which was available on the survey home page. Subsequently, electronic informed consent was requested and obtained from the participants, thus ensuring that they agreed to participate voluntarily in the research. The project was reviewed and approved by the Research Ethics Committee of Universidad Peruana Unión (approval number: 2022-CEUPeU-0009). In addition, permission was obtained from the university's academic vice-rectorate. The study was conducted in accordance with the ethical standards and amendments included in the Declaration of Helsinki.

## Adherence to a healthy lifestyle

To assess adherence to a healthy lifestyle, we used the Diet and Healthy Lifestyle Scale (DEVS) [26], the validated Peruvian version of the Vegetarian Lifestyle Index (VLI), which was originally developed by Le and colleagues [2]. The VLI was developed following the recommendations of the guidelines for healthful diets and lifestyles proposed by the Department of Nutrition, School of Public Health, Loma Linda University [27]. The Peruvian validation, like the VLI, is composed of 14 items, 11 of which include topics related to plant-based diets, considering the consumption of whole foods of plant origin, such as fruits, vegetables, legumes, nuts, seeds, and whole grains. Similarly, foods of animal origin, such as milk and dairy products, eggs, were considered reliable sources of vitamin B-12. Candies (sweets) were also considered. Additionally, the last 3 items represent lifestyle characteristics and include regular physical activity, adequate water intake, and moderate sunlight exposure. For each question, the response options were limited to 3. The 14 items are summed to obtain a total score ranging from 0 to 14 points, considering the following scoring system: 0, 0.5 or 1 point. Participants received a score based on how well they followed the recommendations: They were awarded 1 point for complete compliance, 0.5 points for partial compliance, and 0 points if there was no compliance. For example, participants who consumed ≥6 servings/day of whole grains received a score of 1, those who consumed ≥3 and <6 servings/day received a score of 0.5; and if they consumed less than 3 servings/day of whole grains they received a score of 0. Additionally, a reverse scoring was applied for the following components: vegetable oils, dairy products, eggs, sweets, and flesh-food intake, whose recommendations were to consume in moderation or in moderate amounts, so that higher intake of these foods received lower scores. For example, participants who consumed >5 servings/week of sweets received 0 points. Those who consumed >2 and ≤5 servings/week received 0.5 points. While those who consumed 0 to 2 servings per week received 1 point. On the other hand, in terms of lifestyle variables, we have considered the following: For sunlight exposure: <5 min/day = 0 point (low), ≥5 and <10

min/day = 0.5 points (middle), and $\geq$10 min/day = 1 point (high). Water intake: <4 glasses of water/day = 0 point (low), $\geq$4 and <8 glasses of water/day = 0.5 points (middle), and $\geq$8 glasses of water/day = 1 point (high). Daily exercise: 0 min/day of any moderate or vigorous exercise = 0 point (low), >0 and <30 min/day of moderate exercise or >0 and <15 min/day of vigorous exercise, and $\geq$30 min/day of moderate exercise or $\geq$15 min/day of vigorous exercise = 1 point (high). Higher total scores indicate greater adherence to healthy lifestyle [2, 12].

## Identification of dietary patterns

The dietary patterns of the participants were assessed using the following question, "What type of diet do you follow?". The response options were as follows: (a) vegan (exclusive consumption of plant-based foods), (b) ovo-vegetarian (consumption of plant-based foods, eggs, and derivatives), (c) lacto-vegetarian (consumption of plant-based foods, milk, and dairy products), (d) lacto-ovo-vegetarian (consumption of foods of vegetable origin, eggs, milk, and dairy products), (e) pesco-vegetarian (consumption of plant-based foods, fish, eggs, milk, and dairy products), and (f) non-vegetarian (consumption of plant-based foods, meat, fish, eggs, milk, and dairy products). In addition, the dietary patterns were classified as non-vegetarians, semi-vegetarians, and vegetarians according to the definitions of Le and colleagues (Table 1) [2].

## Sociodemographic information

Information was collected based on 8 items considering the following categories: age (years) (18–20, 21–25, and $\geq$26), sex (male and female), place of origin (coast, highlands, jungle, and foreigners), place of residence (rural and urban), marital status (single and married), and religion (Catholic and Protestant). In addition, we considered the faculty of study (business, engineering, education, and health sciences), and the parents' level of education (elementary, technical, and university).

## BMI

Additionally, information was collected on the weight and height of the students. These data were self-reported. Subsequently, the BMI was calculated and classified according to the parameters established by the WHO: (a) underweight, <18.5 kg/m$^2$; (b) normal weight, 18.5–24.9 kg/m$^2$; (c) overweight, 25.0–29.9 kg/m$^2$; (d) obesity $\geq$30 kg/m$^2$ [28].

**Table 1. Definitions of dietary patterns.**

| Dietary patterns | Meats | Chicken | Fish | Dairy | Eggs | Vegetables, fruits, breads, grains, cereals, oilseeds, and legumes |
|---|---|---|---|---|---|---|
| Non-vegetarian[1] | Yes | Yes | Yes | Yes | Yes | Yes |
| Semi-vegetarian[2] | No | Yes | Yes | Yes | Yes | Yes |
| Vegetarian | | | | | | |
| Pescovegetarian[3] | No | No | Yes | Yes | Yes | Yes |
| Lacto-ovo vegetarian[4] | No | No | No | Yes | Yes | Yes |
| Vegan[5] | No | No | No | No | No | Yes |

Note.

[1] There are no specific dietary restrictions on the frequency of consumption of meat, fish, and dairy products.

[2] They consume red meat, poultry, or fish no more than once a week.

[3] They eat fish, but not red meat or poultry.

[4] They consume milk and dairy products, eggs, and foods of vegetable origin.

[5] Exclusion of all meat and products of animal origin.

## Statistical analysis

Data analysis was performed using the R program version 4.0.2 (R Foundation for Statistical Computing, Vienna, Austria; http://www.R-project.org). Categorical and continuous variables were described as frequency (%) or mean (M) ± standard deviation (SD). For comparative analysis, the chi-square or ANOVA test was performed. To assess the independent association of significant factors with a healthy lifestyle and dietary pattern, odds ratios (ORs) and their respective 95% confidence intervals (95% CI) were determined using bivariate and multivariate logistic regression models. To determine the ORs using logistic regressions, the dependent variable dietary pattern was dichotomized into vegetarian and non-vegetarian (semi-vegetarian and non-vegetarian). A $p < 0.05$ was considered statistically significant in all analyzes.

## Results

Of the 6846 university students evaluated, 2,850 were non-vegetarians, 2,281 were semi-vegetarians, and 1,715 were vegetarians. In Table 2, we found that 52.8% were aged between 18 and 20 years; 54.4% were women, predominantly from the highlands (54.7%) and the largest proportion lived in urban areas (71%). In addition, 95.6% were single students, 69.5% were Protestant, 32.3% and 31.3% were students enrolled in the faculty of health sciences and engineering, respectively, and 69.3% reported adequate BMI. Regarding the educational level of the parents, 56.9% had basic education. According to the distribution of the DEVS, we found that students aged 18 to 20 years presented a higher adherence to healthy lifestyle (6.41±1.46, $p = 0.015$). Likewise, we evidenced statistically significant differences in score distribution in relation to male sex (6.47±1.43, $p < 0.001$), foreign students (6.74±1.50, p = 0.007), students residing in rural areas (6.42±1.43, $p = 0.062$), students enrolled in the faculty of education (6.58±1.54, $p < 0.001$), those whose parents were highly educated (6.46±1.49, $p = 0.04$), vegetarians (6.72 ±1.44, $p < 0.001$), and BMI, evidencing those students with obesity had a significantly lower healthy lifestyle score (5.97±1.48, $p < 0.001$).

Table 3 presents the distribution of sociodemographic characteristics and other selected attributes in relation to the dietary patterns of the participants. This table reveals that students who opted for the non-vegetarian and semi-vegetarian diet were predominantly female and aged between 18 and 20 years. In addition, it was identified that they attended to the faculty of health sciences and came from families whose parents had basic education levels. Furthermore, it is important to mention that more than 25% of the students with excess body weight (overweight and obesity) belonged to this group. In contrast, students who followed a vegetarian dietary pattern were mostly male. On the other hand, students who resided mainly in urban areas in the highlands of Peru and identified themselves as Protestants were predominantly semi-vegetarians and vegetarians. In addition, most of them attended engineering faculty. A relevant characteristic is that a large percentage of these students presented a lower BMI, in the category of normal BMI and underweight.

In Fig 1, we evidenced that 45.7%, 49.8%, and 54.1% of the respondents reported moderate levels of sunlight exposure, water consumption, and physical activity, respectively. Consumption of meats, vitamin B-12, eggs, and dairy products was moderate in 44.1%, 48.7%, 65.8%, and 63.1% of the surveyed population, respectively. However, the consumption of sweets and vegetable oils was high in 57.1% and 66.7% of the participants, respectively. The consumption of plant-based foods such as nuts, fruits, vegetables, legumes, and whole grains was reported to be low in 59%, 35.9%, 46%, 38.5, and 57.7% of the respondents, respectively.

The sociodemographic factors associated with the vegetarian dietary pattern are shown in Table 4. The logistic regression model adjusted for male sex showed that older students (21–25 or ≥26 years), Protestant students and students enrolled in business or education faculties

**Table 2. Distribution of DEVS scores among students according to sociodemographic characteristics, dietary patterns, and BMI.**

| Characteristics | Distribution | | DEVS score | | p-value |
|---|---|---|---|---|---|
| | n | % | Mean | SD | |
| Age (years) | | | | | 0.015* |
| 18–20 | 3618 | 52.8 | 6.41 | 1.46 | α→ε$^{¥}$ |
| 21–25 | 2654 | 38.8 | 6.33 | 1.45 | |
| ≥26 | 574 | 8.38 | 6.26 | 1.51 | |
| Sex | | | | | <0.001$^{++}$ |
| Male | 3119 | 45.6 | 6.47 | 1.43 | |
| Female | 3727 | 54.4 | 6.28 | 1.49 | |
| Location of origin | | | | | 0.007** |
| Coast | 1491 | 21.8 | 6.37 | 1.54 | α→ω$^{¥}$ |
| Highlands | 3748 | 54.7 | 6.36 | 1.42 | ε→ω$^{¥}$ |
| Jungle | 1434 | 20.9 | 6.33 | 1.49 | β→ω$^{¥}$ |
| Foreign student | 173 | 2.53 | 6.74 | 1.5 | |
| Place of residence | | | | | 0.062 |
| Rural | 1963 | 28.7 | 6.42 | 1.43 | |
| Urban | 4883 | 71.3 | 6.35 | 1.47 | |
| Marital status | | | | | 0.232 |
| Single | 6542 | 95.6 | 6.37 | 1.46 | |
| Married | 304 | 4.4 | 6.27 | 1.48 | |
| Religion | | | | | 0.242 |
| Catholic | 2090 | 30.5 | 6.4 | 1.4 | |
| Protestant | 4756 | 69.5 | 6.35 | 1.49 | |
| Faculty of study | | | | | <0.001** |
| Business | 1787 | 26.1 | 6.3 | 1.44 | α→ω$^{¥}$ |
| Engineering | 2143 | 31.3 | 6.3 | 1.41 | β→ω$^{¥}$ |
| Education | 703 | 10.3 | 6.58 | 1.54 | α→ε$^{¥}$ |
| Health Sciences | 2213 | 32.3 | 6.42 | 1.49 | β→ε$^{¥}$ |
| Parents education level | | | | | 0.04* |
| Basic | 3895 | 56.9 | 6.32 | 1.44 | α→ε$^{¥}$ |
| Technical | 1223 | 17.9 | 6.38 | 1.5 | |
| Higher education | 1728 | 25.2 | 6.46 | 1.49 | |
| Dietary pattern | | | | | <0.001** |
| Omnivore | 2850 | 41.6 | 6.04 | 1.4 | α→β$^{¥}$ |
| Semi-vegetarian | 2281 | 33.3 | 6.51 | 1.46 | β→ε$^{¥}$ |
| Vegetarian | 1715 | 25.1 | 6.72 | 1.44 | α→ε$^{¥}$ |
| BMI | | | | | <0.001** |
| Low weight | 179 | 2.6 | 6.42 | 1.53 | α→ω$^{¥}$ |
| Normal | 4745 | 69.3 | 6.43 | 1.44 | β→ω$^{¥}$ |
| Overweight | 1650 | 24.1 | 6.24 | 1.51 | β→ε$^{¥}$ |
| Obesity | 272 | 4 | 5.97 | 1.48 | ε→ω$^{¥}$ |

Note. DEVS, Diet and Healthy Lifestyle Scale; BMI, Body mass index; SD, Standart deviation.

$^{+}$p<0.05 or

$^{++}$p<0.01, statistical significance by T-student;

*p < 0.05 or

**p < 0.01, statistical significance by ANOVA.

$^{¥}$p<0.05, statistical significance by post hoc Tukey test.

**Table 3. Sociodemographic and other selected characteristics of university students according to dietary patterns.**

| Characteristics | Non-vegetarian | | Semi-vegetarian | | Vegetarian | | p-value |
|---|---|---|---|---|---|---|---|
| | n | % | n | % | n | % | |
| Sex | | | | | | | <0.001** |
| Male | 1241 | 43.5 | 959 | 42.0 | 919 | 53.6 | |
| Female | 1609 | 56.5 | 1322 | 58.0 | 796 | 46.4 | |
| Age | | | | | | | <0.001** |
| 18–20 years | 1559 | 54.7 | 1246 | 54.6 | 813 | 47.4 | |
| 21–25 years | 1085 | 38.1 | 836 | 36.7 | 733 | 42.7 | |
| ≥26 years | 206 | 7.2 | 199 | 8.7 | 169 | 9.8 | |
| Location of origin | | | | | | | <0.001** |
| Coast | 647 | 22.7 | 458 | 20.1 | 386 | 22.5 | |
| Highlands | 1489 | 52.2 | 1330 | 58.3 | 929 | 54.2 | |
| Jungle | 649 | 22.8 | 442 | 19.4 | 343 | 20.0 | |
| Foreign student | 65 | 2.3 | 51 | 2.2 | 57 | 3.3 | |
| Place of residence | | | | | | | <0.001** |
| Rural | 631 | 22.1 | 698 | 30.6 | 634 | 37.0 | |
| Urban | 2219 | 77.9 | 1583 | 69.4 | 1081 | 63.0 | |
| Marital status | | | | | | | 0.051 |
| Single | 2731 | 95.8 | 2190 | 96.0 | 1621 | 94.5 | |
| Married | 119 | 4.2 | 91 | 4.0 | 94 | 5.5 | |
| Religion | | | | | | | <0.001** |
| Catholic | 1041 | 36.5 | 625 | 27.4 | 424 | 24.7 | |
| Protestant | 1809 | 63.5 | 1656 | 72.6 | 1291 | 75.3 | |
| Faculty of study | | | | | | | <0.001** |
| Business | 756 | 26.5 | 585 | 25.6 | 446 | 26.0 | |
| Engineering | 912 | 32.0 | 671 | 29.4 | 560 | 32.7 | |
| Education | 227 | 8.0 | 253 | 11.1 | 223 | 13.0 | |
| Health Sciences | 955 | 33.5 | 772 | 33.8 | 486 | 28.3 | |
| Parents education level | | | | | | | <0.001** |
| Basic | 1424 | 50.0 | 1421 | 62.3 | 1050 | 61.3 | |
| Technical | 569 | 20.0 | 380 | 16.7 | 274 | 16.0 | |
| Higher education | 857 | 30.0 | 480 | 21.0 | 391 | 22.7 | |
| BMI | | | | | | | <0.001** |
| Low weight | 79 | 2.8 | 54 | 2.4 | 46 | 2.7 | |
| Normal | 1841 | 64.6 | 1668 | 73.1 | 1236 | 72.1 | |
| Overweight | 771 | 27.1 | 501 | 22.0 | 378 | 22.0 | |
| Obesity | 159 | 5.6 | 58 | 2.5 | 55 | 3.2 | |

Note. BMI, Body mass index;

**p<0.01; statistical significance by Chi-square test.

presented a higher probability of having a vegetarian dietary pattern; however, students residing in urban areas and those with parents with technical studies were less likely to have a vegetarian diet. Similarly, the regression model in females shows that female students who were Protestant and enrolled in the faculty of education were more likely to have a vegetarian diet; however, students from the jungle region and those living in rural areas had a lower probability.

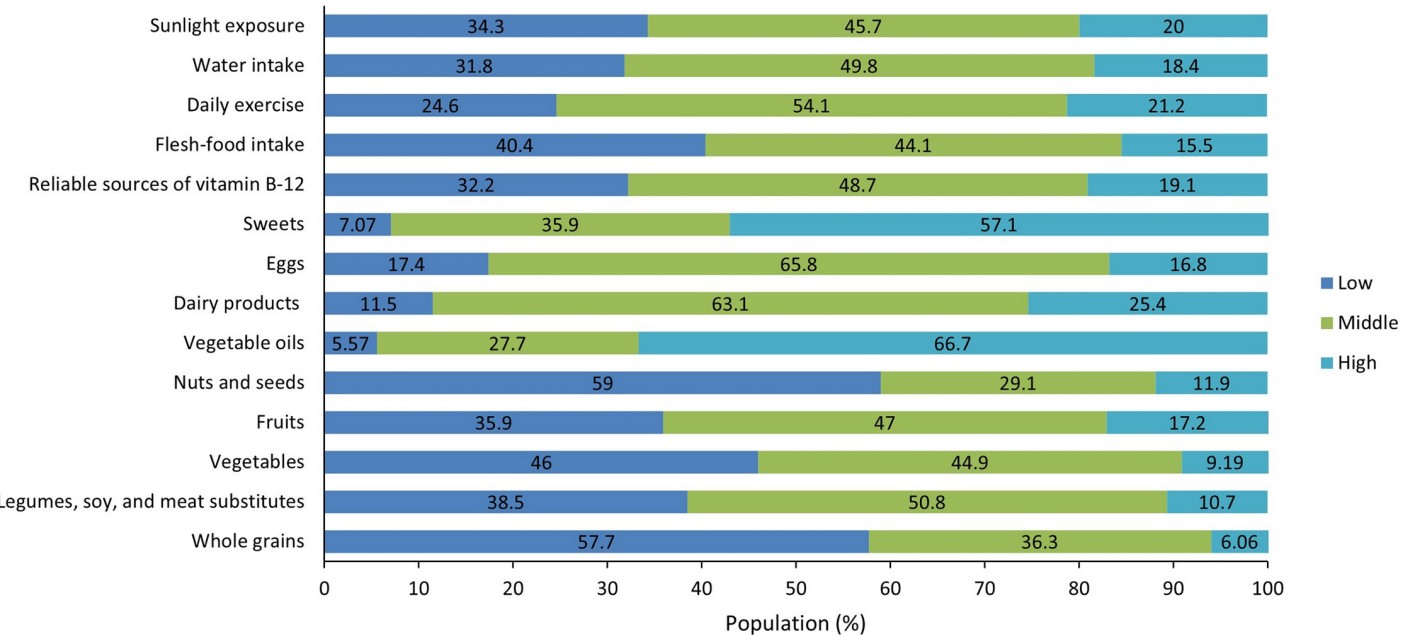

**Fig 1. Distribution of DEVS among Peruvian university students.** Note: Each color line shows the frequency of the lifestyle habit, as level of sunlight exposure, duration or consumption is represented as dark blue for "Low," light green for "Middle," and light blue for "High".

Table 5 presents the sociodemographic and BMI factors associated with the adherence to a healthy lifestyle. The logistic regression model stratified for sex showed that semi-vegetarian, vegetarian (pesco-vegetarian, lacto-ovo-vegetarian, and vegan) students and those from abroad were more likely to have healthy lifestyle. However, those who studied in the faculty of engineering and with excess body weight (overweight and obesity) were less likely. On the other hand, the regression model in females evidenced that semi-vegetarian, vegetarian students, those whose parents had a technical and university level of education, were more likely to report higher adherence to a healthy lifestyle.

## Discussion

The current study expands and provides new information on lifestyle and vegetarianism among university students. Furthermore, it is the first cross-sectional survey of this reach and magnitude in Peru; therefore, it provides an important perspective on the adherence to a healthy lifestyle among Peruvian university students.

The results showed that semi-vegetarians and vegetarians (pesco-vegetarians, lacto-ovo-vegetarians, and vegans), had better adherence to a healthy lifestyle compared to non-vegetarians. In addition, those who self-reported being vegetarian have lower BMI (normal BMI and underweight). Students with excess body weight (overweight and obesity), however, were less likely to report adherence to a healthy lifestyle. On the other hand, the results of the analysis of adherence to a healthy lifestyle for the entire population showed that health-related lifestyle behaviors, such as sunlight exposure, water intake, and regular physical activity, were predominantly moderate and low. Similarly, the intake of sweets was high; while the intake of healthy foods such as fruits, vegetables, legumes, and whole grains was reported to be low.

The results of the study showed that vegetarians were more likely to be male; however, most previous studies conducted in the vegetarian population found that the predominant sex is

**Table 4. Sociodemographic factors associated with the vegetarian dietary pattern in the participants.**

| Factors | Men | | Women | |
|---|---|---|---|---|
| | OR[a] | 95% CI | OR[a] | 95% CI |
| Age | | | | |
| 18–20 years | 1 | reference | 1 | reference |
| 21–25 years | 1.29 | (1.09–1.53)** | 1.14 | 0.96–1.35 |
| ≥26 years | 1.34 | (1.02–1.75)* | 1.27 | 0.92–1.72 |
| Location of origin | | | | |
| Coast | 1 | reference | 1 | reference |
| Highlands | 0.97 | (0.78–1.20) | 0.9 | (0.73–1.10) |
| Jungle | 1.12 | (0.88–1.44) | 0.68 | (0.53–0.88)** |
| Foreign student | 1.16 | (0.71–1.87) | 1.45 | (0.88–2.36) |
| Place of residence | | | | |
| Rural | 1 | reference | 1 | reference |
| Urban | 0.57 | (0.48–0.67)** | 0.66 | (0.56–0.79)** |
| Religion | | | | |
| Catholic | 1 | reference | 1 | reference |
| Protestant | 1.31 | (1.09–1.58)** | 1.36 | (1.13–1.64)** |
| Faculty of study | | | | |
| Health Sciences | 1 | reference | 1 | reference |
| Business | 1.32 | (1.03–1.69)* | 1.04 | (0.85–1.27) |
| Engineering | 1.17 | (0.93–1.47) | 1.1 | (0.88–1.37) |
| Education | 1.54 | (1.14–2.08)** | 1.34 | (1.03–1.74)* |
| Parents education level | | | | |
| Basic | 1 | reference | 1 | reference |
| Technical | 0.77 | (0.61–0.96)* | 0.91 | (0.73–1.13) |
| Higher education | 0.84 | (0.69–1.02) | 0.91 | (0.75–1.12) |

Note. OR[a], adjusted Odds ratio by all factors; 95% CI, 95% Confidence interval;

*p<0.05 or

**p<0.01, statistical significance by logistic regression. To determine odd ratios through logistic regressions in this table, the dependent variable dietary pattern was dichotomized into vegetarian and non-vegetarian (semi-vegetarian and non-vegetarian).

usually female [11, 12, 18, 29]. On the other hand, men were more likely to report a higher adherence score to healthy lifestyle. It is possible that the findings obtained in this study are influenced, at least in part, by the selection of the faculty to which the participants belong. For example, more than 30% of the respondents were found to be part of the engineering faculty. It is important to note that engineering disciplines have historically shown a higher prevalence of male students compared to female students. However, it is relevant to note that in recent years there has been an increase in female participation in this field [30, 31]. Regarding age, it was observed that students following a non-vegetarian or semi-vegetarian diet were observed to range in age from 18 to 20 years of age. These results are consistent with those reported by Olfert and colleagues [18], who found that most 18-year-old students followed non-vegetarian diets. However, other studies report contrary results, showing that vegetarians were younger than their non-vegetarian counterparts [29, 32]. For example, a study conducted in the Canadian population found that the tendency to consume less animal foods is more common among younger age groups [33]. This phenomenon could be attributed to growing awareness among young people of the importance of environmental protection and animal welfare [34].

**Table 5. Factors associated with adherence to a healthy lifestyle in university students.**

| Factors | Men | | Women | |
|---|---|---|---|---|
| | OR[a] | 95% CI | OR[a] | 95% CI |
| Dietary pattern | | | | |
| Omnivore | 1 | reference | 1 | reference |
| Semi-vegetarian | 1.66 | (1.36–2.03)** | 2.07 | (1.73–2.48)** |
| Vegetarian | 1.78 | (1.45–2.17)** | 2.63 | (2.16–3.22)** |
| Age | | | | |
| 18–20 years | 1 | reference | 1 | reference |
| 21–25 years | 0.98 | (0.82–1.16) | 0.78 | (0.66–0.92)** |
| ≥26 years | 0.96 | (0.72–1.26) | 0.79 | (0.57–1.08) |
| Location of origin | | | | |
| Coast | 1 | reference | 1 | reference |
| Highlands | 0.81 | (0.66–1.00)* | 0.93 | (0.76–1.13) |
| Jungle | 0.99 | (0.77–1.26) | 0.96 | (0.76–1.22) |
| Foreign student | 1.85 | (1.16–2.93)** | 1.19 | (0.71–1.95) |
| Faculty of study | | | | |
| Health Sciences | 1 | reference | 1 | reference |
| Business | 0.9 | 0.70–1.15 | 0.91 | 0.75–1.09 |
| Engineering | 0.73 | (0.59–0.91)** | 0.93 | 0.75–1.15 |
| Education | 1.24 | 0.92–1.67 | 1.03 | 0.79–1.33 |
| Parents education level | | | | |
| Basic | 1 | reference | 1 | reference |
| Technical | 0.93 | 0.74–1.17 | 1.45 | (1.18–1.77)** |
| Higher education | 1.15 | 0.94–1.39 | 1.38 | (1.15–1.67)** |
| BMI | | | | |
| Normal | 1 | reference | 1 | reference |
| Low weight | 0.91 | (0.56–1.45) | 0.83 | (0.49–1.36) |
| Overweight | 0.81 | (0.65–0.97)* | 0.76 | (0.63–0.91)** |
| Obesity | 0.55 | (0.34–0.85)** | 0.77 | (0.48–1.19) |

Note. BMI, Body mass index; OR[a], adjusted Odds ratio by all factors; 95% CI, 95% Confidence interval;

*p<0.05 or

**p<0.01, statistical significance by logistic regression.

Living in rural areas could have positive effects on adopting a healthy lifestyle, including a higher consumption of fresh food and a lower risk of sedentary behaviors [35]. In this study, although most students from urban areas showed a preference for a vegetarian diet, it is interesting to note that those from rural areas showed a greater propensity to adopt a healthy lifestyle. In fact, students living in rural areas had a better score of adherence to a healthy lifestyle compared to students living in urban areas. Our findings are consistent with the results of a study showing that participants living in rural areas were more physically active and consumed more fruits, while urbanization was associated with a sedentary lifestyle, higher consumption of high-fat foods, and low consumption of fruits and vegetables [36]. University students living in urban areas could be exposed to important changes in dietary patterns and have a greater inclination towards sedentary behaviors, due to factors such as the influence of the media (food marketing), nutritional transition, availability of new technological tools, and globalization [37]. This pace of urban life replaces traditional diets, characterized by fishing and

agriculture, giving rise to a consumption pattern defined by an increased intake of ultra-processed foods [38]. Therefore, future research should focus on the development of interventions based on university students considering the areas of residence to improve public awareness of the importance of leading a healthy lifestyle.

Interestingly, Protestant students were mostly vegetarian; furthermore, according to the logistic regression analysis, being Protestant represented a probability of being vegetarian. The university where the research was conducted belongs to the Seventh-day Adventist Church (SDA), an organization that works to promote healthy lifestyle, including vegetarian diets [39]. It is speculated that a large portion of the students surveyed are members of the SDA church. The SDA church established hundreds of institutions such as hospitals, universities, high schools, and tens of thousands of churches around the world, all promoting the vegetarian lifestyle among patients, teachers, students, and followers [40]. These factors could explain the reasons for these findings. SDA Church members are known for their healthy lifestyle, including the practice of a plant-based diet [41].

In the current study, we found that semi-vegetarians and vegetarians (pesco-vegetarians, lacto-ovo-vegetarians, and vegans), had a better healthy lifestyle score compared to non-vegetarians. These findings are like the results reported in studies conducted in the vegetarian population. For example, a study that evaluated healthy lifestyles found that vegetarians reported healthier lifestyle habits and lower risk factors for noncommunicable diseases than non-vegetarians [12]; in addition, this same study reported that vegans had a higher mean adherence to the healthy lifestyle [12]. Likewise, Olfert et al. [18]. in a study of vegetarian and non-vegetarian university students found that vegetarians show adequate dietary patterns (e.g., higher consumption of fruits and vegetables, lower percentage of energy obtained from fat consumption). These results are consistent with the findings reported by Le and colleagues [2], who reported that vegetarians tended to report higher scores of adherence to a healthy lifestyle than their non-vegetarian counterparts. The observed variety in lifestyle adherence scores between vegetarian and non-vegetarian university students could be since vegetarian diets tend to favor the consumption of minimally processed plant-based foods, such as whole grains and meat substitutes, and natural foods such as fruits, vegetables, legumes, soybeans, nuts, and seeds [34, 42]. Generally, in most of the studies conducted, the analyzes indicated that vegetarians obtain higher index scores for healthy eating and lifestyle [12, 18, 43, 44].

Vegetarian diets have gained recognition and popularity as one of the most healthful dietary patterns [45]. The popularity of vegetarian diets is mainly related to their nutritional and health benefits associated with a lower risk of non-communicable diseases and a better anthropometric profile, including a normal BMI [11, 42, 46]. Current studies have suggested that vegetarian diets are essentially anti-obesogenic compared with diets that include meat and derived products [47]. In the current study, vegetarian diets (pesco-vegetarian, lacto-ovo-vegetarian, and vegan) showed a significantly higher proportion among students with lower BMI (normal weight and underweight). These findings are supported by the results of previous studies conducted in Peruvian vegetarians and non-vegetarians [11, 34], which concluded that vegetarians had a lower BMI compared to non-vegetarians. Furthermore, Chiu et al. [48], in a sample of Taiwanese vegetarians and a Brazilian study [46], found that the BMI of non-vegetarians was >27.0 kg/m2 and >24.9 kg/m2, respectively, compared to vegetarians. The possible justification behind these results could be the fact that the diets are characterized by a lower caloric density, due to the presence of a higher content of dietary fiber and bioactive elements (phytochemicals) [49]. The relationship between vegetarian diets and adequate BMI is well documented in the scientific literature. Several studies have shown that vegetarianism is significantly associated with lower BMI, total cholesterol concentration, LDL-cholesterol,

glucose levels, systolic blood pressure, diastolic blood pressure blood glucose levels [11, 34, 50, 51]. and a lower risk of all-cause mortality [44].

In contrast, diets based on the consumption of animal foods are characterized by high caloric density, excess saturated and trans fats, sodium, and added sugars [52]. In fact, regular consumption of these foods can contribute to serious alterations in the anthropometric and lipid profile, such as elevated BMI, waist circumference and fat percentage, increased concentration of triglycerides, low-density lipoproteins, and glucose, which, in turn, increases cardiometabolic risk factors, obesity, hypertension, diabetes mellitus, and cardiovascular disease [34, 53]. In the present study, we found that students with excess body weight (overweight and obesity) were less likely to report adherence to healthy lifestyle. Similarly, in another study, BMI increased with a greater inclusion of meat and meat products and/or fish in the diet [54]. Therefore, it is important to encourage the adoption of a vegetarian diet in the university population to prevent non-communicable diseases, such as obesity and other diseases associated with food based on the consumption of meat and meat products.

It should be mentioned that excluding foods of animal origin from the diet does not necessarily guarantee a healthy dietary pattern or optimal health, but, it also depends on other lifestyle elements, such as physical activity, adequate rest, sunlight exposure, and adequate water consumption [11]. In the young population, for both sexes, unhealthy lifestyle patterns tend to be the highest [55]. In the present investigation, the results of the analysis of adherence to a healthy lifestyle for the entire population showed that health-related lifestyle behaviors, such as sunlight exposure, water consumption, and regular physical activity, were predominantly moderate and low. A recent study by Almutairi and colleagues [25] conducted in university students reported a low level of adherence to recommendations on physical activity and healthy eating habits. Similarly, other previous studies have reported that health-related behaviors such as physical activity and sunlight exposure were low or intermediate [12]. Several studies have shown that lack of physical activity is associated with an increased risk of morbidity and mortality from non-communicable diseases such as obesity, cancer, among others [56]. In addition, insufficient sunlight exposure affects vitamin D supply, especially when there is no compensation through food intake or fortified nutritional supplements [57, 58]. This, in turn, may contribute to a higher incidence of breast cancer, colorectal cancer, cardiovascular disease, type 1 diabetes mellitus, hypertension, metabolic syndrome, autism, multiple sclerosis, Alzheimer's disease, asthma, and myopia [59]. The overall analysis of the findings of the current study evidences the need to design and implement healthy lifestyle programs in vegetarian and non-vegetarian university students to encourage them to be more responsible in adopting health-related behaviors such as physical exercise and healthy eating habits.

The vegetarian dietary pattern is more than abstaining from the consumption of meat and derived products, it consists of the consumption of healthy, plant-based, and minimally processed foods [34]. Following the same order of ideas, some studies suggest that although plant-based foods are related to a better state of health, they are not synonymous with the characteristics of a healthy eating pattern when they are not adequately consumed [60]. In our study, the intake of healthy foods such as fruits, vegetables, legumes, and whole grains was reported to be low. Our findings complement the results of Gili et al., [12] who in a similar study conducted among Argentine vegetarians and non-vegetarians concluded that the levels of adherence to whole foods of plant origin are insufficient, even for vegans. Moreover, Gili y colleagues [12], reported that the consumption of legumes, vegetables, nuts, and seeds was not adequately consumed and, in fact, only 5% of the respondents showed an adequate intake of legumes and vegetables according to the proposed standards for a healthy diet [27]. Inadequate consumption of healthy foods in vegetarians should be the subject of future studies to evaluate the associated consequences.

Another finding to mention is that study participants reported a high intake of sweets. Previous studies have shown that vegetarian diets composed mainly of processed foods may not represent a protective factor for noncommunicable diseases, specifically for coronary heart disease [10]. Likewise, a study conducted in Brazil reported a high frequency of daily consumption of sweet foods, such as sugary drinks and ultra-processed foods [61]. Another study found that greater avoidance of animal foods was associated with greater consumption of ultra-processed foods in pesco-vegetarians, vegetarians, and vegans [62]. However, several previous studies have shown that vegetarians, particularly vegans, consume a limited amount of sugary foods, such as sweets, candies, chocolate, cakes, and carbonated and sweetened beverages [63, 64]. The possible justification for the findings of our study could be since the study was conducted in a young population (university students), who tend to consume more sweets and fast food, and less whole grains, fruits, and vegetables [65].

The university environment is a critical space in which students tend to engage in risky health behaviors, which can negatively affect physical health and emotional well-being [25]. The university stage influences lifestyle choices, including eating habits, levels of physical activity, and rest, conditions that indirectly increase the risk of developing noncommunicable diseases such as obesity, especially during the first year of the study [66, 67]. Adhering to a healthy lifestyle on a university campus can be difficult due to the conditions and characteristics of the environment [18]. In addition, vegetarians of university age are at increased risk for disordered eating behaviors, such as binge eating, intentional vomiting, and laxative use [68, 69]. Given that existing problems in university students, the impact of vegetarian diets on young adults' eating behaviors needs further investigation. In Peru, universities should implement intervention strategies and policies in the university environment to promote vegetarian diets aligned with established dietary guidelines according to the criteria of a vegetarian diet. In the United States, universities have been very active with students in terms of encouraging healthy eating. In fact, 80% of U.S. campuses have vegetarian options in university restaurants [70]. In addition, some universities have made efforts to improve student health by promoting semi-vegetarian and vegetarian diets through strategies such as the "Meatless Monday" campaign [71]. The promotion of healthy lifestyles and food consumption by universities is particularly essential to reduce the prevalence of overweight and obesity among their students.

Finally, it is important to mention that vegetarian diets, although they have many known health benefits, may also be associated with certain nutritional deficiencies, particularly vitamins B12 and D, iron, calcium, zinc, and omega-3 fatty acids [8]. These nutrients are essential for the maintenance of general health and well-being, and their deficiency can significantly impact quality of life and the ability to maintain a healthy lifestyle [9]. Nutritional deficiencies and healthy lifestyle habits in vegetarian university students are a critical aspect to consider [72]. For example, iron deficiency can lead to lower energy and concentration, which could negatively influence physical activity and academic performance, key aspects of a healthy lifestyle [9]. In addition, a lack of essential vitamins could impact mental and emotional health, important factors for adherence to a healthy lifestyle [73].

## Strengths and limitations

The strengths of the current study consist of a large sample size (more than 6,000 university students). In addition, to our knowledge, this is the first study on the adherence to a healthy lifestyle conducted among vegetarian and non-vegetarian university students in Peru. However, the results should be interpreted considering certain limitations. First, this is a cross-sectional study and, therefore, its interpretation is limited to the temporality of cause and effect. Second, the results of the study cannot be generalized to all Peruvian university students

because its sampling technique was limited to a single university, although it has campuses in the three main regions of the country (coast, jungle, and highlands). Third, all information collected was self-reported; therefore, the definitions of "non-vegetarian", semi-vegetarian", and "vegetarian" may not be accurate because they were not based on dietary intake. Fourth, we do not know exactly the proportion of vegetarian subgroups (pesco-vegetarians, lacto-vegetarians, lacto-ovo-vegetarians, and vegans) present in the study. It is possible that the results could have been affected by the lack of this information; therefore, further studies focusing on university students are warranted to explore the behavior of various vegetarian subgroups compared to the general vegetarian population. Finally, other aspects of lifestyle habits, such as alcohol consumption and smoking, were not evaluated.

## Conclusions

The main findings showed that semi-vegetarian and vegetarian students had a better score for adherence to a healthy lifestyle. In addition, students who exhibited lower BMI values (underweight and normal weight) were those who opted for a vegetarian diet. However, given that, in all participants, the levels of adherence to a health-related lifestyle (sunlight exposure, water consumption, and regular physical activity) and the consumption of healthy plant-based foods were insufficient, the need for future research on this topic in this population group becomes more evident.

Factors such as access to plant-based foods, the cost of vegetarian foods, their preparation, and the time to adhere to the vegetarian diet should be included in future research. Furthermore, considering that the university environment does not favor adherence to plant-based diets, universities should implement strategies and policies for interventions in the university environment to promote vegetarian diets aligned to established dietary guidelines according to the criteria for a healthy diet. Finally, the promotion of a healthy lifestyle by higher education institutions is essentially necessary to reduce the prevalence of overweight and obesity among their students.

## Acknowledgments

We thank the subjects for agreeing to participate in the study. We would also like to thank Dr. Varisier Noel for his support in the research development process.

## Author Contributions

**Conceptualization:** Jacksaint Saintila, Edda E. Newball-Noriega, Salomón Huancahuire-Vega, Yaquelin E. Calizaya-Milla.

**Data curation:** Ricardo Rojas-Humpire.

**Formal analysis:** Ricardo Rojas-Humpire.

**Funding acquisition:** Jacksaint Saintila, Salomón Huancahuire-Vega.

**Investigation:** Jacksaint Saintila.

**Methodology:** Jacksaint Saintila, Felipe L. Ignacio-Cconchoy.

**Project administration:** Yaquelin E. Calizaya-Milla.

**Supervision:** Salomón Huancahuire-Vega, Felipe L. Ignacio-Cconchoy.

**Validation:** Edda E. Newball-Noriega, Salomón Huancahuire-Vega.

**Visualization:** Edda E. Newball-Noriega.

**Writing – original draft:** Jacksaint Saintila, Salomón Huancahuire-Vega, Yaquelin E. Calizaya-Milla.

**Writing – review & editing:** Jacksaint Saintila, Ricardo Rojas-Humpire, Edda E. Newball-Noriega, Salomón Huancahuire-Vega, Felipe L. Ignacio-Cconchoy, Yaquelin E. Calizaya-Milla.

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
