## [Decision Letter · Decision Letter 0]

25 Jan 2024

PONE-D-23-38350Analysis of adherence to a healthy lifestyle among vegetarian and non-vegetarian Peruvian university students: A cross-sectional surveyPLOS ONE

Dear Dr. Saintila,

Thank you for submitting your manuscript to PLOS ONE. After careful consideration, we feel that it has merit but does not fully meet PLOS ONE’s publication criteria as it currently stands. Therefore, we invite you to submit a revised version of the manuscript that addresses the points raised during the review process.

We look forward to receiving your revised manuscript.

Kind regards,

Melissa Orlandin Premaor, M.D., Ph.D

Academic Editor

PLOS ONE

Journal Requirements:

"The study was financed by the Universidad Peruana Unión, Peru (Resolución N° 2556-2022/UPeU-CU)."

3. We note that your Data Availability Statement is currently as follows: All relevant data are within the paper and its Supporting Information files.

Reviewers' comments:

Reviewer's Responses to Questions

**Comments to the Author**

1. Is the manuscript technically sound, and do the data support the conclusions?

Reviewer #1: Yes

Reviewer #2: Yes

2. Has the statistical analysis been performed appropriately and rigorously? 

Reviewer #1: Yes

Reviewer #2: Yes

3. Have the authors made all data underlying the findings in their manuscript fully available?

Reviewer #1: Yes

Reviewer #2: Yes

4. Is the manuscript presented in an intelligible fashion and written in standard English?

Reviewer #1: Yes

Reviewer #2: Yes

5. Review Comments to the Author

Reviewer #1: I propose to address the following issues.

Introduction:

The author needs to put some specific rationale, why they chose to compare between vegetarian and nonvegetarian.

Methodology:

Line: 110: With self-reported height and weight, the chance of bias or error is high due to flat slope-syndrome. The author needs to mention how they ensured the accuracy of the height and weight.

Results:

Line 216: Table 2: Under the table the author mentioned the P value is from ANOVA. However, for categorical variables with two groups, like sex, the test should be T Test instead of ANOVA. Moreover, for ANOVA out its better to mention the pairwise comparison, rather than just to mention the overall P value.

Discussion:

The discussion part is written is right order. However, in my opinion, it may also add the credibility of the article if the add few information on deficiency of nutrients, the vegetarian diet flower could face, and is it linked to the outcome of the study.

Figure 1: In my opinion the author needs to change the color pattern of Fig 1 to make it clear for the reader.

Reviewer #2: Dear authors

It was a pleasure going over your manuscript. Please find below recommendations and few edits for improvement.

Before line 70, please include a paragraph on the adherence to mediterranean diet and kindly include some of the references below.

Karam, J., Serhan, C., Swaidan, E., & Serhan, M. (2022). Comparative Study Regarding

the Adherence to the Mediterranean Diet Among Older Adults Living in Lebanon and

Syria. Frontiers in Nutrition, 9, 972. https://doi.org/10.3389/FNUT.2022.893963/BIBTEX

Karam, J., Ghach, W., Bouteen, C., Makary, M.-J., Riman, M., & Serhan, M. (2022).

Adherence to Mediterranean diet among adults during the COVID-19 outbreak and the

economic crisis in Lebanon. Nutrition & Food Science, 52(6), 1018–1028.

https://doi.org/10.1108/NFS-10-2021-0325

Karam, J., Bibiloni, M. D. M., Serhan, M., & Tur, J. A. (2021). Adherence to

Mediterranean Diet among Lebanese University Students. Nutrients 2021, Vol. 13, Page

1264, 13(4), 1264. https://doi.org/10.3390/NU13041264

6. PLOS authors have the option to publish the peer review history of their article (what does this mean?). If published, this will include your full peer review and any attached files.

Reviewer #1: **Yes: **Md Kamruzzaman

Reviewer #2: No

---

## [Author Response · Author response to Decision Letter 0]

6 Feb 2024

5-02-2024

Dear Dr Emily Chenette

Editor-in-chief

PLOS ONE

Thank you very much for your letter. We have reviewed the manuscript entitled “Analysis of adherence to a healthy lifestyle among vegetarian and non-vegetarian Peruvian university students: A cross-sectional survey”, and we would like to resubmit it for consideration in your journal. We have addressed the comments raised by the reviewers, and the amendments are highlighted in the revised manuscript. We have responded point by point to each reviewer's comments and suggestions.

We hope that the revised version of the manuscript is now acceptable for publication in your journal. In addition, we are very grateful for the contribution of the reviewers in improving the scientific quality of the study.

Sincerely,

Jacksaint Saintila

Reviewer #1: I propose to address the following issues.

Introduction:

The author needs to put some specific rationale, why they chose to compare between vegetarian and nonvegetarian.

R.- Thank you very much for reviewing our manuscript. The choice to compare the lifestyles of vegetarian and nonvegetarian university students in our study is based on several key considerations. First, vegetarian diets are gaining popularity, especially among young people, due to their perception as a healthier, more sustainable, and ethical option compared to traditional meat-based diets.

Second, there is growing interest in understanding how dietary choices, particularly the adoption of a vegetarian diet, affect overall health and well-being. Studying these differences can provide valuable information on the benefits and potential challenges of following a vegetarian diet compared to a non-vegetarian diet, especially at a stage of life where long-term eating and lifestyle patterns are being established.

Therefore, comparison between vegetarians and non-vegetarians is critical to understand not only the direct health effects of these diets, but also to inform individuals, educational institutions, and policy makers on how to best support the health and well-being of youth in a context of dietary diversity. Please review lines 103-114.

Methodology:

Line: 110: With self-reported height and weight, the chance of bias or error is high due to flat slope-syndrome. The author needs to mention how they ensured the accuracy of the height and weight.

R.- Thank you for your comment. It is true that self-reporting of height and weight can carry certain risks of bias or error, such as flat slope-syndrome, where heavier individuals tend to underestimate their weight and shorter individuals tend to overestimate their height. To minimize these risks in our study, we implemented several strategies.

First, at the beginning of the study, participants were emphasized the importance of providing accurate information, assuring them of confidentiality and explaining how their data would contribute significantly to the research. This approach sought to increase accountability and accuracy in their responses.

In addition, we provide detailed and standardized instructions on how to measure height and weight. These instructions included recommendations for using appropriate equipment and advice on how best to perform the measurements (such as being barefoot and fasting for weight, and standing upright against a wall to measure height).

Although we recognize that self-reporting is not as accurate as direct measurements by health professionals, in the context of our study, a large sample and the geographic distribution of participants made direct measurements infeasible. Therefore, this method was chosen as the most feasible and practical, while measures were implemented to improve the accuracy of the self-reported data. Please see lines 117-120.

Results:

Line 216: Table 2: Under the table the author mentioned the P value is from ANOVA. However, for categorical variables with two groups, like sex, the test should be T Test instead of ANOVA. Moreover, for ANOVA out its better to mention the pairwise comparison, rather than just to mention the overall P value.

R.- Thank you very much for your valuable recommendations.

Regarding your comment on Table 2, we wish to clarify that a Chi-square analysis was not applied because the categorical variables are only being described in this section.

We have taken your suggestion into account and a clarification has already been made in the manuscript regarding the variables that are compared by Student's t-test. Additionally, in response to your comment, we have incorporated Tukey's post hoc analysis to facilitate pairwise comparison within the ANOVA. This allows a more detailed interpretation of the previously mentioned global P-values.

All comparatives have been clearly specified with symbols in the table, and this information has been adequately detailed in the legend to ensure an accurate understanding of the data presented.

Discussion:

The discussion part is written is right order. However, in my opinion, it may also add the credibility of the article if the add few information on deficiency of nutrients, the vegetarian diet flower could face, and is it linked to the outcome of the study.

R.- Thank you very much for your recommendation. We have added this aspect on lines 477-487 of the discussion.

Figure 1: In my opinion the author needs to change the color pattern of Fig 1 to make it clear for the reader.

R.- Thank you very much for your recommendation. We have improved the presentation of Figure 1.

 

Reviewer #2: Dear authors

It was a pleasure going over your manuscript. Please find below recommendations and few edits for improvement.

Before line 70, please include a paragraph on the adherence to mediterranean diet and kindly include some of the references below.

• Karam, J., Serhan, C., Swaidan, E., & Serhan, M. (2022). Comparative Study Regarding

the Adherence to the Mediterranean Diet Among Older Adults Living in Lebanon and

Syria. Frontiers in Nutrition, 9, 972. https://doi.org/10.3389/FNUT.2022.893963/BIBTEX

• Karam, J., Ghach, W., Bouteen, C., Makary, M.-J., Riman, M., & Serhan, M. (2022).

Adherence to Mediterranean diet among adults during the COVID-19 outbreak and the

economic crisis in Lebanon. Nutrition & Food Science, 52(6), 1018–1028.

https://doi.org/10.1108/NFS-10-2021-0325

• Karam, J., Bibiloni, M. D. M., Serhan, M., & Tur, J. A. (2021). Adherence to

Mediterranean Diet among Lebanese University Students. Nutrients 2021, Vol. 13, Page

1264, 13(4), 1264. https://doi.org/10.3390/NU13041264

R.- Thank you very much for your recommendation. We have added a new paragraph, establishing a link between vegetarian diets and the Mediterranean diet in terms of their health benefits and their relevance to nutrition education, thus respecting your valuable suggestion without deviating significantly from the central theme of your article.

---

## [Editor Report · Decision Letter 1]

9 Feb 2024

Analysis of adherence to a healthy lifestyle among vegetarian and non-vegetarian Peruvian university students: A cross-sectional survey

PONE-D-23-38350R1

Dear Dr.  Jacksaint Saintila,  

We’re pleased to inform you that your manuscript has been judged scientifically suitable for publication and will be formally accepted for publication once it meets all outstanding technical requirements.

Kind regards,

Melissa Orlandin Premaor, M.D., Ph.D

Academic Editor

PLOS ONE
---

## [Editor Report · Acceptance letter]

12 Feb 2024

PONE-D-23-38350R1 

PLOS ONE

Dear Dr. Saintila, 

I'm pleased to inform you that your manuscript has been deemed suitable for publication in PLOS ONE. Congratulations! Your manuscript is now being handed over to our production team.

Kind regards, 

on behalf of

Dr. Melissa Orlandin Premaor 

Academic Editor

PLOS ONE